# Temporal Sentence Grounding with Relevance Feedback in Videos

**Jianfeng Dong[1 2]**    **Xiaoman Peng[1]**    **Daizong Liu[3]\***    **Xiaoye Qu[4]**

**Xun Yang[5]**    **Cuizhu Bao[1 2]\***    **Meng Wang[6]**

[1]Zhejiang Gongshang University
[2]Zhejiang Key Laboratory of Big Data and Future E-Commerce Technology
[3]Peking University [4]Huazhong University of Science and Technology
[5]University of Science and Technology of China [6]Hefei University of Technology

## Abstract

As a widely explored multi-modal task, Temporal Sentence Grounding in videos (TSG) endeavors to retrieve a specific video segment matched with a given query text from a video. The traditional paradigm for TSG generally assumes that relevant segments always exist within a given video. However, this assumption is restrictive and unrealistic in real-world applications where the existence of a query-related segment is uncertain, easily resulting in erroneous grounding. Motivated by the research gap and practical application, this paper introduces a new task, named Temporal Sentence Grounding with Relevance Feedback (TSG-RF) in videos, which accommodates the possibility that a video may or may not include a segment related to the query. This task entails localizing precise video segments that semantically align with the query text when such content is present, while delivering definitive feedback on the non-existence of related segments when absent. Moreover, we propose a novel Relation-aware Temporal Sentence Grounding (RaTSG) network for addressing this challenging task. This network first reformulates the TSG-RF task as a foreground-background detection problem by investigating whether the query-related semantics exist in both frame and video levels. Then, a multi-granularity relevance discriminator is exploited to produce precise video-query relevance feedback and a relation-aware segment grounding module is employed to selectively conduct the grounding process, dynamically adapting to the presence or absence of query-related segments in videos. To validate our RaTSG network, we reconstruct two popular TSG datasets, establishing a rigorous benchmark for TSG-RF. Experimental results demonstrate the effectiveness of our proposed RaTSG for the TSG-RF task. Our source code is available at https://github.com/HuiGuanLab/RaTSG.

## 1  Introduction

Grounding target content described by users in videos is a fundamental capability that facilitates various multimedia applications, such as intelligent robotic service [1], video on demand [2], and metaverse [3]. Following this demand, temporal sentence grounding in videos (TSG) [4] has recently become a research hotspot, attracting wide attention from researchers in various fields [5–9]. Despite the significant advancements made in this field, the existing TSG task is overly idealistic. It aims to identify segments that are semantically relevant to a given query from a given long video, assuming that relevant segments always exist in the given video [10–12]. However, this setting limits the applicability of TSG in real-world scenarios as the given video may not contain the query-related contents, resulting mismatched or wrong grounding results. As illustrated in Figure 1(a), given a

---

*Corresponding authors.

38th Conference on Neural Information Processing Systems (NeurIPS 2024).

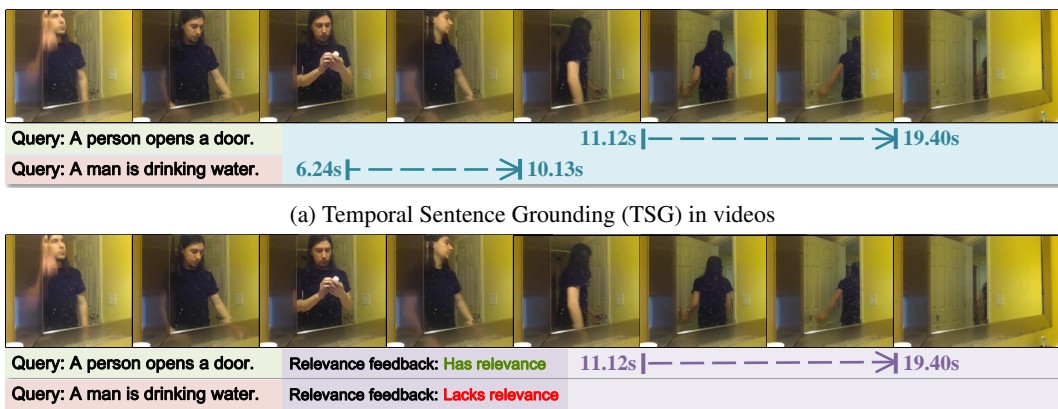

(a) Temporal Sentence Grounding (TSG) in videos

(b) Temporal Sentence Grounding with Relevance Feedback (TSG-RF) in videos

Figure 1: The difference between the TSG and TSG-RF tasks. TSG *always* predicts the start and end boundaries of the grounded segments, even in the absence of video content relevant to the given query text. By contrast, TSG-RF provides relevance feedback on whether exist query-related content in the given video, and *selectively* predicts the start and end boundaries of the grounded segments, according to the presence or absence of query-related segments in videos.

video and a query, existing TSG methods always predict the start and end times of segments, even when the video lacks content relevant to the given query text. This suggests a significant gap between the literature and the real world.

To fill the gap, this paper makes the first attempt to introduce an expanded but challenging task for TSG, named Temporal Sentence Grounding with Relevance Feedback (TSG-RF) in videos. This new task aims to deliver more flexible grounding results for a given query and a video. Specifically, as illustrated in Figure 1(b), if relevant segments are present in the video, the task is to pinpoint the video segment that is semantically relevant to the given query description. Conversely, if no relevant segments exist, it should be explicitly conveyed to the user that "there are no target segments related to the query in the given video, and thus no corresponding segments can be localized."

To tackle this new task, it is necessary to first review the failure reason of traditional TSG methods [13–16]. We argue the essential issue of previous TSG methods is their inability to discern whether grounding results are warranted based on the presence or absence of relevant segments. To alleviate this issue, a straightforward approach to address the TSG-RF task involves a two-stage process: Firstly, a relevance discriminator is trained separately to assess the relevance between the query text and the video, providing relevant feedback. Then, query-video samples identified as relevant by the discriminator are processed using models designed for the TSG task to predict the target segment. Despite the structural simplicity of this integrated approach, it requires training two distinct models, leading to the redundant consumption of computational resources, increased storage demands, and suboptimal overall inference efficiency. Therefore, how to design an end-to-end relevance-aware TSG-RF method is an emerging problem.

Based on the above considerations, we propose to address the TSG-RF task by developing a multi-task learning model, *i.e*, the Relation-aware Temporal Sentence Grounding (RaTSG) network. This model promotes mutual enhancement between relevance discrimination and video segment grounding through shared knowledge, thus facilitating a unified and efficient approach to both tasks. As the essence of relevance discrimination lies in accurately measuring the semantic partial relevance between the query text and the video, we propose to first explore the measurement of partial semantic relevance between text and video at two different granularities: fine-grained and coarse-grained. Specifically, to capture fine-grained partial relevance, we adopt the concept of multiple instance learning [17–19], treating the entire video as a bag with each frame as an instance. Following [20], a video is considered partially related to the text if it contains at least one foreground frame; if all frames are background and irrelevant, the video is deemed unrelated to the text. Considering the semantic incompleteness of individual video frames, a coarse-grained perspective is also adopted to learn the relevance between the global video-level semantics and the full query text. Then, a multi-granularity relevance discriminator is designed to integrate the discriminative capabilities of both granularities to predict the final relevance score. As for the selective grounding process based on the relevance score,

existing segment grounding heads always make grounding for any query-video input, even when no relevant segments are present in the video, which is not suitable for the TSG-RF task. Therefore, we design a relation-aware segment grounding module that takes the previously predicted relevance information of each query-video pair as prior knowledge, and selectively conducts grounding.

We find a recent concurrent work [21] that shares a similar motivation with us. However, it simply treats the irrelevant video contents from an open-set perspective and implicitly utilizes cross-dataset evaluation. Instead, we implement original in-domain data to explicitly learn the potential irrelevant relationship. Further, existing TSG datasets can not be directly used to evaluate our proposed network for TSG-RF, since the datasets assume that a given query text can always localize a corresponding relevant segment from the given video. Hence, we further reconstruct two commonly used datasets, Charades-STA [22] and ActivityNet Captions [23], by including samples without grounding results where the video does not contain the content relevant to the query.

The main contributions of this paper are summarized as follows:

- We formalize a novel TSG task to explore more flexible segment grounding in videos, *i.e.*, Temporal Sentence Grounding with Relevance Feedback (TSG-RF) in videos, which advances user-specified TSG towards more practical applications.

- Targeting the TSG-RF task, we propose a novel Relation-aware Temporal Sentence Grounding (RaTSG) network, which mainly depends on a multi-granularity relevance discriminator and a relation-aware segment grounding module. The multi-granularity relevance discriminator is devised to predict the relevance feedback based on fine-grained and coarse-grained relevance between query text and videos, and the relation-aware segment grounding module selectively predicts the start and end boundaries of the grounded segment.

- To facilitate the evaluation of the TSG-RF task, we reconstruct two commonly used TSG datasets and establish appropriate performance evaluation metrics to meet the setting of TSG-RF. Extensive experiments conducted on these reconstructed datasets demonstrate the effectiveness of the proposed model.

## 2 Related Work

### 2.1 Temporal Sentence Grounding and Highlight Detection in Videos

The Temporal Sentence Grounding in Videos (TSG) task aims to retrieve video segments that match user-input natural language descriptions. Recent efforts for TSG can be typically grouped into proposal-based and proposal-free methods. Proposal-based methods [22, 24–39] follow a two-stage paradigm: generating proposals by dividing the video into clips and then aligning textual semantics with visual features. Proposal-free methods [40–44, 10, 13, 45] offer an end-to-end paradigm, predicting target segments without proposals, enhancing computational efficiency. Although efficient, these methods struggled with segment-level feature capture. To combine strengths, Xiao et al. [45] used an anchor-free approach to generate candidates, then matched them with query statements using anchor-based methods. However, the above methods cannot be directly applied to the TSG-RF task proposed in this paper. They will still predict the start and end times of segments and output retrieved segments, even in the absence of video content relevant to the query text.

Highlight detection task is similar to TSG, which mainly aims to identify the most interesting or important segments within a video based on a given natural language query, focusing on segments that are salient or engaging [46–49]. Recent works typically tackle this task using multi-modal inputs, advanced transformers, or large-scale pretraining techniques. For instance, Moment-DETR [46] introduce a transformer-based approach that simplifies highlight detection by treating it as a set prediction task, eliminating traditional proposal steps. QD-DETR [48] enhances video-text understanding with cross-attention and negative pairs to improve relevance predictions. UniVTG [49] unifies diverse temporal annotations and enables large-scale pretraining to improve generalization across video grounding tasks. Notably, similar to TSG, highlight detection also assumes that every video contains highlights. In contrast, our proposed TSG-RF task not only grounds relevant segments but also accounts for cases where no relevant segments are present.

## 2.2 Cross-Modal Semantic Similarity Learning

This paper extends TSG by incorporating relevance feedback to discern the semantic connection between text and video, introducing techniques for learning cross-modal semantic similarity [50–53]. The prevailing approach aligns text and video semantics by learning a shared space, using a metric to assess similarity. For instance, Miech et al. [50] respectively represent text and video into a feature vector, and compute their similarity via cosine similarity metric. Croitoru et al. [52] use distillation techniques to combine knowledge from multiple pretrained models, enhancing cross-modal similarity learning. Current mainstream methods involve learning multiple shared spaces, with relevance computed as a weighted sum of similarities across these spaces. Li et al. [54] employ a multi-space, multi-loss learning framework, enhancing cross-modal similarity by leveraging complementary encoders. While these techniques assess the whole text-video similarity, TSG-RF requires learning partial relevance, where the text relates to only some frames within the video. For non-retrievable samples, the text is irrelevant to any frame. Therefore, the focus is on learning partial relevance.

# 3 The Proposed Method

## 3.1 Problem Definition and Overview

**TSG-RF task.** Given a video $V = [v_1, v_2, \ldots, v_n]$ and a query text $Q = [q_1, q_2, \ldots, q_m]$ as inputs, the TSG-RF task is asked to predict relevance feedback on whether exist query-related content in the given video. Besides, if the query has relevance with the video, it is asked to conduct grounding to identify the precise boundary indices of video segments that are semantically consistent with the query text in the video $V$. Otherwise, the grounding results are ignored. Compared to the traditional TSG task that always predicts grounding results for all videos, this TSG-RF task is more challenging as it requires not only distinguishing the videos containing and not containing query-relevant segments but also generating explicit non-existence signals for non-relevant videos besides the grounding results.

**Overall framework.** Figure 2 illustrates the framework of our proposed RaTSG network for the TSG-RF task. Given an input video $V$ and a query text $Q$ encoded by the attention-based mechanisms, the multi-granularity relevance discriminator first captures fine-grained and coarse-grained semantic-aware relevance between the video and query text at the frame and segment levels. Subsequently, it integrates the relevance determination capabilities of these two granularities to generate the relevance discrimination score $P_v$ for distinguishing whether the video has a semantic-related content of the query. Then, a relation-aware segment grounding module takes the multimodal enhanced video features as the basic representation for generating the grounding probability distribution $P_s$ of the start boundary and $P_e$ of the end boundary, while exploiting relevance feedback signal as additional input to produce the separate final results of irrelevant and relevant segments. In the following sections, we will provide detailed illustration of each component.

## 3.2 Preprocessing of Text and Video Features

Given an input video and a query text, we first use a pre-trained video feature extractor, such as I3D [55], to extract the video feature sequence $V = [v_1, v_2, \ldots, v_n] \in \mathbb{R}^{n \times d_v}$, where $d_v$ is the feature dimension. Subsequently, the query text features are extracted using a text feature extractor, like GloVe [56], producing word embeddings $Q = [q_1, q_2, \ldots, q_m] \in \mathbb{R}^{m \times d_q}$, where $d_q$ represents the feature dimension. Both the video feature sequence $V$ and the query text embeddings $Q$ are first mapped into the same dimension $d$ through a fully connected layer respectively, resulting in $V \in \mathbb{R}^{n \times d}$ and $Q \in \mathbb{R}^{m \times d}$. To enhance the video semantic with the query contexts, we utilize an attention mechanism to integrate word-wise query features $Q$ with the frame-wise video feature $V$ for generating text-guided enhanced video feature $V_q$ [13]. To comprehend the whole semantic of the query for latter global-level reasoning, we use a self-attention layer to perform the intra-modal information interaction on the text feature sequences $Q$ to obtain the sentence-level query representation $h_Q$. The final contextual multi-modal representation $V_q'$ can be obtained by concatenating the frame-wise video feature $V_q$ with sentence-level query feature $h_Q$. More details can be found in Appendix A.1.

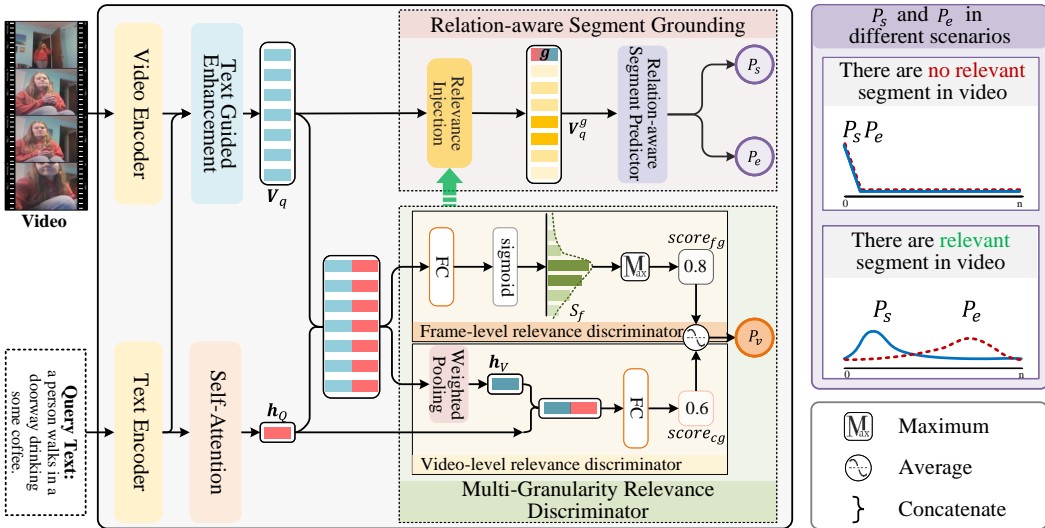

Figure 2: The overall framework of our proposed RaTSG for addressing TSG-RF task. It mainly depends on a multi-granularity relevance discriminator that is employed to learn query-video relevance, and a relation-aware segment grounding module that is used to selectively perform grounding.

## 3.3 Multi-Granularity Relevance Discriminator

Before generating the grounding results for each query-video pair, we need to first investigate the semantic relevance between them. To determine whether a given video contains segments relevant to a query text, the key lies in assessing the existence of relevance between the text semantic and the video content. Here, we delve into capturing the multi-grained query-video relevance by matching the global sentence-level textual contexts with frame-level and video-level video semantics, respectively.

**Frame-level relevance discriminator.** Determining frame-level query-video relevance can be treated as a foreground-background frame discrimination problem. Theoretically, for a video containing query-related content, the foreground frames located within the start and end index boundaries should exhibit strong relevance to the query text. Conversely, for a video lacking relevant segments, no foreground frames in the video should relate to the text. To detect foreground frames from the query text, we design a foreground frame detector that finely learns the similarity between the text and each video frame. Specifically, we feed the fused multi-modal feature sequence $V_q'$ into a fully connected layer to predict a sequence of query-to-frame similarity scores. These scores are then mapped to a range between 0 and 1 using the sigmoid function, resulting in the foreground frame prediction score sequence $S_f = \text{sigmoid}(\text{FC}(V_q')) \in \mathbb{R}^n$ where each score indicates the confidence of a frame belonging to the foreground. To train the frame-level relevance discriminator, a binary cross-entropy loss $L_{frame}$ (Appendix A.2) is employed on each frame, which encourages the discriminator to output large scores for foreground frames while small scores for background frames.

To obtain the relevance between text and video at a fine-grained frame-level, we apply the concept of multiple instance learning [17–19]. To be specific, the video is treated as a bag with video frames as instances, categorized into foreground (positive) and background (negative) frames. Thus, a video containing at least one foreground frame is considered a positive bag, indicating partial relevance to the query text. In contrast, a video composed entirely of background frames is deemed a negative bag, showing no relevance. Following [20], we utilize a max-pooling operator to transfer the instance scores to bag scores, obtaining the fine-grained query-video discrimination score as $score_{fg} = \max(S_f)$. The low fine-grained discrimination score suggests that there are no foreground frames in the video, while a high score indicates the presence of at least one foreground frame.

**Video-level relevance discriminator.** In addition to capturing the frame-level relevance, we also attempt to perceive the semantic relevance from a more informationally complete video level. To generate the comprehensive video-level visual representation, we need to aggregate all query-related frame-wise features into a global one to determine its semantic similarity with the sentence contexts. Specifically, we apply a frame-wise weighted sum operation on the text-guided enhanced video feature $V_q$ to obtain the query-related video representation $h_V \in \mathbb{R}^d$, where we assign weights to

each frame by normalizing the previously obtained frame-level similarity scores (reflect the relative relevance score between the text and each frame) using the softmax function. Then, we fuse the sentence-level textual feature $h_Q$ with the global video feature $h_v$ to obtain the video-level relation signal vector $g$, which is then input into a fully connected layer with a sigmoid function to obtain the final video-level similarity score $score_{cg} = \text{sigmoid}(\text{FC}(g))$, which globally describes the relevance between the text and the entire video. To train the video-level relevance discriminator, a binary cross-entropy loss $L_{video}$ (Appendix A.3) is employed on each query-video sample, which encourages the discriminator output large scores for samples with grounding results while small scores for that without grounding results.

**Multi-level relevance prediction.** The previously discussed two relevance discriminators assess the relevance between query and video at the frame and video levels respectively, yielding fine-grained relevance score ($score_{fg}$) and coarse-grained relevance score ($score_{cg}$). Considering that both granularity levels are deemed equally important for determining the relevance between text and video, we calculate the average of both fine-grained and coarse-grained relevance scores as the final multi-granularity relevance score $P_v$ as:

$$P_v = (score_{fg} + score_{cg})/2. \tag{1}$$

## 3.4 Relation-Aware Segment Grounding

After obtaining the query-video relevance score, we aim to design a flexible segment grounding module to selectively predict the start and end boundaries of the target segments, according to the presence or absence of query-related segments in videos. However, existing segment grounding heads designed for TSG tasks [13, 15, 57, 16] are not suitable for the TSG-RF task. They always generate high-confidence start/end boundaries of segments even when there are no relevant segments in the video. To this end, we introduce a relation-aware segment grounding module that incorporates previously obtained relevance feedback during the segment grounding process, which accommodates the possibility that there may be no relevant segments within the video.

Specifically, to address situations where the video contains no relevant segments, we prepend a special token to the general video feature sequence, representing index 0. This token serves as both the starting and ending boundary indices for samples that do not have grounding results, indicating that the query text does not align with any start and end boundaries in the given video. Therefore, it is essential to introduce a contextual relevance-aware representation as the special feature token to ensure that the segment predictor can effectively handle the absence of relevant segments in the video. Considering the video-level relation signal vector $g$ obtained from the video-level relevance discriminator contains enough relevance-aware knowledge to determine whether the segment exists or not, we utilize the video-level relation signal vector $g$ as the special token in this segment predictor. Consequently, the range of boundary indices for relevant segment grounding is $[1, n]$. Thus, as shown in Figure 2, if a sample is determined to lack relevant segments, the ground-truth boundary index is assigned as $A = [0, 0]$; if a sample is deemed to have grounding results, the ground-truth boundary index is defined as $A = [a^s, a^e]$, where $1 \leq a^s < a^e \leq n$.

To predict the probability distributions of start and end boundaries, following [13], we use two unidirectional LSTMs with two feed-forward layers. The boundary prediction loss function $L_{boundary}$ is represented as:

$$L_{boundary} = -\frac{1}{2}(\sum Y_s \log(P_s) + \sum Y_e \log(P_e)), \tag{2}$$

where $Y_s$ and $Y_e$ are the one-hot vector representations of the ground-truth start ($a_s$) and end ($a_e$) boundary indices, respectively. $P_s$ and $P_e$ indicate the predicted start and end boundary probability distribution, respectively.

## 3.5 Training and Inference

In summary, the total training loss can be defined as:

$$L_{total} = L_{boundary} + \beta L_{frame} + \gamma L_{video}, \tag{3}$$

where $\beta$ and $\gamma$ are hyperparameters used to balance the importance of the three losses.

During the inference phase, the multi-granularity relevance score $P_v$ is first compared with a threshold $m = 0.5$ to determine whether query-related content exists or not in the given video:

$$\begin{cases} \text{Has grounding result,} & \text{if } P_v \geq m \\ \text{No grounding result,} & \text{if } P_v < m \end{cases}. \tag{4}$$

For samples judged to have grounding results, the model predicts the start $(\hat{a}_s)$ and end $(\hat{a}_e)$ boundary indices of the target segment. Specifically, we first compute the joint probability distribution according to the start and the end boundary probability distributions $P_s, P_e$ generated by the relation-aware segment grounding module. The predicted start $(\hat{a}_s)$ and end $(\hat{a}_e)$ boundary indices are obtained by maximizing the joint probability:

$$(\hat{a}_s, \hat{a}_e) = \arg \max_{\hat{a}_s, \hat{a}_e} (P_s^T P_e), \quad 0 \leq \hat{a}_s \leq \hat{a}_e \leq n. \tag{5}$$

## 4 Evaluation

### 4.1 Experimental Setup

**Datasets.** To verify the viability of our proposed model for TSG-RF, samples without grounding results where the video does not contain the content relevant to the query are required. Therefore, we reconstruct two commonly used datasets, Charades-STA [22] and ActivityNet Captions [23], by adding samples without grounding results (Details of how we obtain them are illustrated in the Appendix A.5.). For each sample in both validation and test sets, we add a corresponding sample without grounding result, resulting in 1:1 ratio of samples with and without grounding results.

The Charades-STA [22] dataset comprises 6,672 videos, where the training set contains 12,408 video-text sample pairs, while the test set comprises 3,720 pairs. Since the original Charades-STA does not have a validation set, we randomly halve the original test samples to form a validation set and a test set. The ActivityNet Captions [23] dataset consists of approximately 20,000 videos featuring diverse and open-content videos. Consistent with previous work [43] for the TSG task, the training set includes 37,421 video-text pairs, while the validation and test sets contain 17,505 and 17,031 samples, respectively. After reconstruction, the validation and test sets were augmented with an equal number of samples without grounding results, doubling the total number of sample pairs to 35,010 and 34,062, respectively.

For ease of reference, we name the corresponding reconstructed datasets as *Charades-RF* and *ActivityNet-RF*, respectively.

**Performance metric.** As our proposed TSG-RF task requires models to provide relevance feedback indicating samples with or without grounding results, we use accuracy ("Acc") to measure the ability of relevance feedback. For measuring the grounding ability, referring to the previous TSG works [13, 43], we utilize the "R{n}@{m}" and "mIoU". "R{n}@{m}" indicates the percentage of query texts in the top-*n* segments where at least one instance has an Intersection over Union (IoU) greater than *m*. "mIoU" represents the mean IoU across all test samples. It is worth noting that due to the inclusion of samples with no grounding results, we redefine the calculation of Intersection over Union (IoU) in four specific scenarios: (1) When the sample is predicted to have no grounding results but actually has grounding results, the IoU is set to 0. (2) When both the prediction and the ground truth indicate that the sample has no grounding results, the IoU is set to 1. (3) When the sample is predicted to have grounding results but actually has no grounding results, the IoU is set to 0. (4) When both the prediction and the ground truth indicate that the sample has grounding results, the IoU is calculated as the Intersection over Union between the predicted segment and the ground truth.

**Implementation details.** The implementation details are presented in Appendix A.6.

### 4.2 Comparison with Baseline Methods

As models specifically designed for TSG-RF are non-existing, we compare our proposed model with models targeted at conventional TSG, and adapt them to TSG-RF by adding an extra relevance discriminator for relevance feedback. Specifically, we select six traditional TSG models, including VSLNet [13], SeqPAN [14], EAMAT [15], ADPN [16], UniVTG [49], QD-DETR [48] considering their source code are released thus ensuring fair and replicable comparisons. For the relevance

Table 1: Performance comparison on Charades-RF and ActivityNet-RF dataset. Model$^{++}$ denotes that the baseline model adapted to TRF-RF by utilizing an additional trained relevance discriminator, using two cascaded models: a relevance discrimination model and a segment grounding model. Our proposed RaTSG, a unified model for both relevance discrimination and segment grounding, achieves the best performance with a very lightweight model for TRF-RF.

| Method | Charades-RF | | | | | ActivityNet-RF | | | | | Params (M) |
| --- | --- | --- | --- | --- | --- | --- | --- | --- | --- | --- | --- |
| | Acc | R1@0.3 | R1@0.5 | R1@0.7 | mIoU | Acc | R1@0.3 | R1@0.5 | R1@0.7 | mIoU | |
| VSLNet | 50.00 | 33.74 | 27.31 | 17.72 | 24.69 | 50.00 | 31.06 | 21.88 | 12.82 | 22.27 | **1.16** |
| UniVTG | 50.00 | 35.81 | 30.03 | 16.67 | 24.96 | 50.00 | 30.89 | 21.67 | 11.29 | 21.35 | 41.35 |
| QD-DETR | 50.00 | 35.16 | 29.46 | 19.27 | 25.31 | 50.00 | 26.50 | 19.15 | 11.07 | 18.99 | 7.07 |
| ADPN | 50.00 | 35.62 | 28.44 | 19.87 | 25.98 | 50.00 | 30.72 | 20.74 | 12.38 | 22.05 | 2.27 |
| SeqPAN | 50.00 | 35.35 | 29.57 | 20.51 | 26.14 | 50.00 | 31.85 | 22.65 | 13.34 | 22.86 | 1.19 |
| EAMAT | 50.00 | 37.12 | 30.59 | 20.86 | 27.27 | 50.00 | 31.10 | 20.80 | 12.07 | 22.07 | 94.12 |
| VSLNet$^{++}$ | 71.94 | 61.40 | 56.77 | 49.65 | 54.67 | 81.60 | 66.15 | 58.37 | 50.64 | 58.65 | 5.34 |
| UniVTG$^{++}$ | 71.94 | 62.58 | 58.55 | 48.79 | 54.65 | 81.60 | 66.15 | 58.36 | 49.46 | 58.00 | 45.53 |
| QD-DETR$^{++}$ | 71.94 | 62.18 | 58.20 | 50.96 | 55.13 | 81.60 | 62.43 | 56.13 | 49.27 | 55.97 | 11.25 |
| ADPN$^{++}$ | 71.94 | 62.26 | 57.23 | 51.16 | 55.41 | 81.60 | 65.85 | 57.41 | 50.28 | 58.47 | 6.45 |
| SeqPAN$^{++}$ | 71.94 | 62.12 | 58.01 | 51.61 | 55.49 | 81.60 | 66.77 | 58.98 | 51.11 | 59.11 | 5.37 |
| EAMAT$^{++}$ | 71.94 | 63.55 | 59.17 | 51.96 | 56.23 | 81.60 | 66.13 | 57.36 | 49.93 | 58.45 | 98.30 |
| **RaTSG (ours)** | **76.85** | **68.17** | **61.91** | **54.19** | **59.93** | **84.27** | **69.02** | **60.68** | **52.88** | **61.15** | 1.27 |

discriminator, we utilize a binary classification model which is trained separately to assess the relevance between the query text and the video (For the details of the relevance discriminator, please refer to Appendix B.1). Then, query-video samples identified as relevant by the discriminator are processed using models designed for the TSG task to predict the target segment. For ease of reference, we denote the corresponding models associated with an extra relevance discriminator as VSLNet$^{++}$, SeqPAN$^{++}$, EAMAT$^{++}$, ADPN$^{++}$, UniVTG$^{++}$, and QD-DETR$^{++}$ respectively.

Table 1 summarizes the performance comparison and model parameters on the Charades-RF and ActivityNet-RF dataset. It is worth noting that traditional TSG models lack the ability to discriminate relevance. These models assume all samples have grounding results, making them incapable of correctly handling samples without grounding results. Hence, this often leads to mismatched grounding prediction, resulting in low recall and mIoU performance for TSG-RF. Additionally, since the test set has an equal ratio (1:1) of samples with and without grounding results, the relevance prediction accuracy of these models is only 50%. Additionally, the enhanced versions of these baseline models, namely VSLNet$^{++}$, SeqPAN$^{++}$, EAMAT$^{++}$, ADPN$^{++}$, UniVTG$^{++}$, and QD-DETR$^{++}$, include a relevance discriminator, which results in consistent performance gains compared to their counterparts without the discriminator. However, these enhanced models require separate and independent training of the relevance discriminator and video grounding components, leading to redundant use of computational resources and increased model size. In contrast, our proposed RaTSG model provides a more lightweight and comprehensive solution by integrating the discrimination and grounding modules, achieving the best performance.

## 4.3 Ablation Studies

### 4.3.1 Effectiveness of the Multi-Granularity Relevance Discriminator

To validate the effectiveness of the multi-granularity relevance discriminator, we compare it with degraded models that use only coarse-grained or fine-grained discrimination scores. As shown in Table 2, single-granularity models fail to perceive the various degrees of partial relevance between text and video, therefore performing worse than the multi-granularity one.

### 4.3.2 Effectiveness of the Relation-aware Segment Grounding

To assess the viability of the relation-aware segment grounding module, we conduct ablations on it by replacing the relation signal vector $g$ with a randomly initialized one. As shown in Table 3, the relation-aware segment grounding consistently outperforms the random counterpart. The result demonstrates the benefit of using previously predicted relevance information of each query-video pair as prior knowledge for TSG-RF. Additionally, in order to further explore how relation features are

Table 2: The effectiveness of the multi-granularity discriminator on Charades-RF.

| Granularity | | Acc | R1@0.3 | R1@0.5 | R1@0.7 | mIoU |
|---|---|---|---|---|---|---|
| coarse | fine | | | | | |
| ✗ | ✓ | 75.35 | 67.34 | 60.91 | 53.84 | 59.27 |
| ✓ | ✗ | 75.73 | 67.63 | 60.48 | 53.25 | 59.18 |
| ✓ | ✓ | **76.85** | **68.17** | **61.91** | **54.19** | **59.93** |

Table 3: The effectiveness of the relation-aware segment grounding on Charades-RF.

| Relation-aware | Acc | R1@0.3 | R1@0.5 | R1@0.7 | mIoU |
|---|---|---|---|---|---|
| ✗ | 76.40 | 66.18 | 59.62 | 51.96 | 57.82 |
| ✓ | **76.85** | **68.17** | **61.91** | **54.19** | **59.93** |

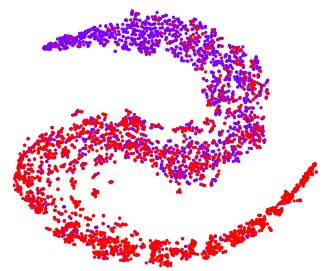

Figure 3: t-SNE visualization of relation signal vectors of all test samples on Charades-RF, where red dots represent relevant samples and blue points represent irrelevant samples of queries and videos.

Table 4: The performance comparison of relevance discrimination with and without the segment grounding module on Charades-RF.

| Segment Grounding | Acc |
|---|---|
| ✗ | 75.59 |
| ✓ | **76.85** |

Table 5: The performance comparison of segment grounding with and without the relevance discriminator on Charades-STA.

| Discriminator | R1@0.3 | R1@0.5 | R1@0.7 | mIoU |
|---|---|---|---|---|
| ✗ | 67.47 | 54.62 | 35.43 | 49.37 |
| ✓ | **74.19** | **56.61** | **37.47** | **53.02** |

Table 6: Our proposed multi-granularity relevance discriminator and relation-aware segment grounding enhance can be jointly used to enhance traditional TSG methods for TSG-RF tasks.

| Method | Acc | R1@0.3 | R1@0.5 | R1@0.7 | mIoU |
|---|---|---|---|---|---|
| EAMAT | 50.00 | 37.12 | 30.59 | 20.86 | 27.27 |
| EAMAT[++] | 71.94 | 63.55 | 59.17 | 51.96 | 56.23 |
| **EAMAT+Ours** | **76.37** | **67.47** | **62.02** | **54.83** | **59.58** |

learned, we visualize the relation signal feature $g$ by t-SNE [58] in Figure 3. It is demonstrated that dots with the same color are relatively more clustered than those with different colors. Overall, the result shows the good discrimination ability of the learned relation feature in our model for relevant and irrelevant samples.

### 4.3.3 Mutual Enhancement between Relevance Discrimination and Segment Grounding

Our proposed RaTSG is a unified model for both relevance discrimination and segment grounding tasks, trained in a multi-task learning manner. To explore whether the two tasks mutually enhance each other, we evaluate the counterparts with the corresponding segment grounding module or relevance discriminator removed. As shown in Table 4, removing the segment grounding module influences the performance of relevance discrimination. Similarly, as shown in Table 5, removing the relevance discriminator degrades the performance of segment grounding. The results allow us to conclude that the relevance discriminator and the segment grounding in our model mutually enhance each other, which also demonstrates the effectiveness of our unified dual-branch framework design.

### 4.3.4 Enhancing traditional TSG methods to work for TSG-RF tasks.

To further investigate the scalability of our proposed multi-granularity relevance discriminator and relation-aware segment grounding modules, we also conduct experiments to explore whether they can be adapted for traditional TSG methods to enable them to work for the TSG-RF task. Specifically, we integrate these components into an existing traditional TSG model, EAMAT, for comparison. As shown in Table 6, EAMAT with our devised multi-granularity relevance discriminator and relation-aware segment grounding outperforms the original EAMAT and the enhanced EAMAT[++] with an extra relevance discriminator. These results demonstrate the adaptability and effectiveness of our proposed modules for enhancing traditional TSG methods to work for the TSG-RF task.

## 4.4 Analysis of Grounding Examples

Figure 4 visualizes grounding examples obtained by our proposed RaTSG and VSLNet++. The results show that our RaTSG effectively handles samples with and without grounding results. For the first sample with grounding results, RaTSG localizes video segments more accurately than VSLNet++. Furthermore, foreground frame prediction scores obtained by RaTSG are more reasonable. We attribute it to the fact that RaTSG is trained using samples without grounding results, allowing the model to learn the similarity characteristics between background frames and the text, thus enhancing its ability to distinguish foreground and background frames in videos.

For the second sample without grounding results in Figure 4, RaTSG consistently predicts low foreground frame scores, providing accurate relevance feedback. Besides, the relation-aware segment grounding module assigns a high probability of the special index of 0, indicating no grounding result. In contrast, VSLNet++ provides incorrect relevance feedback for this sample, showing high foreground frame prediction scores and resulting in incorrect grounding segments.

In Figure 5, we illustrate two bad examples to discuss the limitations of our model. In the first example, our model incorrectly judges the relevance feedback due to the lack of audio cues which are crucial for identifying the action of sneezing. In the second example, our model misinterprets the temporal sequence of actions, mistaking the closing action for the opening action. These examples demonstrate that our proposed model struggles with handling audio-related actions and temporal-sensitive content. However, such limitation can be alleviated by integrating audio features and temporal modeling in the video representation moudel.

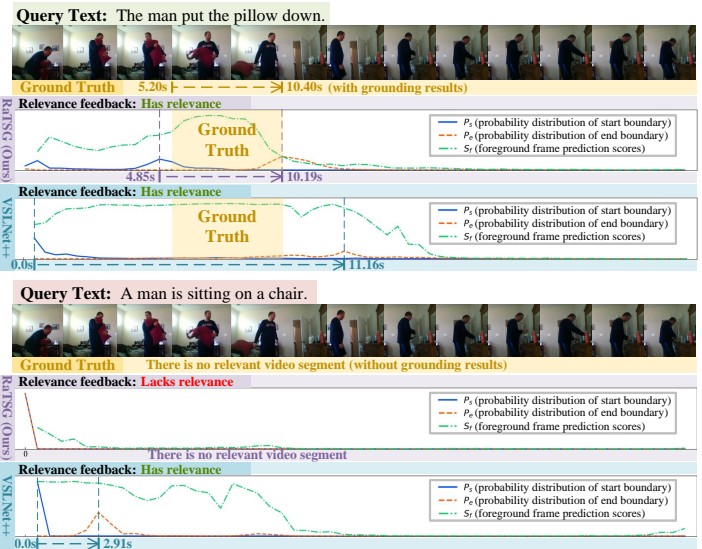

Figure 4: Visualization of grounding examples obtained by our proposed RaTSG and the VSLNet++ baseline.

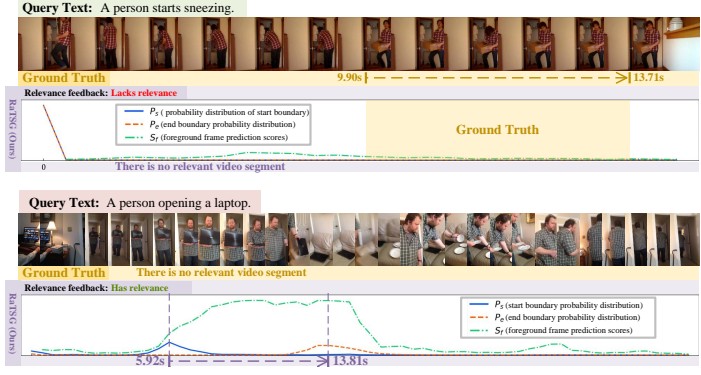

Figure 5: Visualization of two bad examples obtained by our proposed RaTSG.

## 5 Conclusions

This paper breaks through the bottleneck of existing TSG works that can not handle the cases where query-related segments do not exist, by introducing a more realistic and natural task, *i.e.*, TSG-RF. By incorporating the multi-granularity relevance discriminator into the segment grounding model with further mutual enhancement designs, our proposed method can effectively and efficiently localize precise video segments that closely match the query text for videos containing relevant content, while providing clear feedback indicating the absence of related segments for videos not containing relevant content. Besides, two constructed datasets for TSG-RF are contributed.

## Acknowledgements

This work was supported by the Pioneer and Leading Goose R&D Program of Zhejiang (No. 2024C01110, No. 2023C01212), Young Elite Scientists Sponsorship Program by China Association for Science and Technology (No. 2022QNRC001), Zhejiang Provincial Natural Science Foundation (No. LZ23F020004), Fundamental Research Funds for the Provincial Universities of Zhejiang (No. FR2402ZD) and the Graduate Research Innovation Fund Project of Zhejiang Gongshang University.

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

# A  More Technical Details

## A.1  Video and Text Encoders

Considering the need for intra-modal interaction among features, this paper implements a single layer of the simplest Transformer encoder to achieve self-attention. Since the Transformer primarily models global information interactions and does not capture local region information within the feature sequence, four convolutional layers are added prior to the multi-head attention mechanism in the Transformer to enhance the capability for local area interaction.

### A.1.1  Self-Attention Mechanism

To capture the complete semantic information and obtain sentence-level features of the query text, a self-attention mechanism is employed to aggregate the word embedding sequence. Specifically, the sentence-level feature $h_Q$ of the query text can be calculated with the word-wise query feature $Q$ as follows:

$$h_Q = Q^T \cdot \text{softmax}(QW), \tag{6}$$

where $h_Q \in \mathbb{R}^d$, and $W \in \mathbb{R}^{d \times 1}$ is a learnable weight matrix that represents the importance of each word in the sentence.

### A.1.2  Text-Guided Video Feature Enhancement

To enhance the video semantic with the query contexts, we employ a query-video bidirectional attention mechanism [59, 60], which allows for simultaneous focus on the interaction between text words and video frames. The query-video bidirectional attention mechanism calculates attention scores from two directions: from video to text and from text to video. Both directions of attention matrices are derived from a common similarity matrix $S \in \mathbb{R}^{n \times m}$, which can be expressed as:

$$s_{ij} = \alpha(V_i, Q_j), \tag{7}$$

where $s_{ij} \in \mathbb{R}$ represents the similarity score between the $i$-th video feature vector $V_i$ and the $j$-th word vector $Q_j$. The function $\alpha(\cdot)$ is a trilinear function [61] used to measure the similarity between the two input vectors $V_i$ and $Q_j$:

$$\alpha(v, q) = W_0[v; q; v \odot q], \tag{8}$$

where $W_0$ represents a trainable weight matrix, and $\odot$ denotes element-wise multiplication. Normalizing the rows of the common similarity matrix $S$ yields the attention weights from video to text $A^{v2q} = [a_1^{v2q}, a_2^{v2q}, \ldots, a_n^{v2q}] \in \mathbb{R}^{n \times m}$. Similarly, normalizing the columns of $S$ gives the attention weights from text to video $A^{q2v} = [a_1^{q2v}, a_2^{q2v}, \ldots, a_m^{q2v}] \in \mathbb{R}^{m \times n}$. Subsequently, using the attention weights $A^{v2q}$, the weighted text feature sequence $Q$ is used to construct the video feature sequence $C^v = A^{v2q} \cdot Q \in \mathbb{R}^{n \times d}$. Similarly, utilizing the attention weights $A^{q2v}$, the text feature sequence reconstructed from video features $C^{q2v} = A^{q2v} \cdot V' \in \mathbb{R}^{m \times d}$ is obtained. The reconstructed text feature sequence $C^{q2v}$ is then mapped back into the video encoding space using the attention weights $A^{v2q}$, producing a text-aware video feature sequence $C^q = A^{v2q} \cdot C^{q2v} \in \mathbb{R}^{n \times d}$. Finally, through a simple feedforward network, the original video feature sequence $V$, the video feature sequence reconstructed from text features $C^v$, and the text-aware video feature sequence $C^q$ are fused to obtain the text-guided enhanced video feature $V_q$:

$$V_q = \text{FFN}([V; C^v; V \odot C^v; V \odot C^q]), \tag{9}$$

where $V_q \in \mathbb{R}^{n \times d}$; FFN$(\cdot)$ denotes the feedforward network; $\odot$ indicates element-wise multiplication.

### A.1.3  Multi-Modal Feature Fusion

Although the bidirectional attention mechanism facilitates cross-modal interaction and information fusion between query text and video, this process primarily enables local interaction between individual words and video frames. Therefore, we further integrate sentence-level features $h_Q$, which contain complete semantic information, with the text-guided enhanced video feature $V_q$ to generate the final contextual multi-modal representation $V_q'$. This is achieved through the feature concatenation operation:

$$V_q' = \text{FC}([h_Q, v_{q1}, h_Q; v_{q2}, \ldots, h_Q; v_{qn}]), \tag{10}$$

where $[;]$ denotes concatenation along the feature dimension, and FC$(\cdot)$ represents a fully connected layer that maps the features to a $d$-dimensional space.

## A.2  Supervision of the Frame-level Relevance Discriminator

Considering that the similarity score between text and each video frame depends on the fully connected layer's understanding of foreground and background frames, this paper introduces a binary cross-entropy loss $L_{frame}$ to

supervise the learning of the rrame-level relevance discriminator. This supervision ensures that the discriminator correctly learns the differences between foreground frames and background frames. The loss function is defined as follows:

$$L_{frame} = -\frac{1}{n}\sum_{i=1}^{n}\left(y_{f_i}\log(s_{f_i}) + (1 - y_{f_i})\log(1 - s_{f_i})\right),\tag{11}$$

where $y_{f_i}$ is the $i$-th element of the binary sequence $Y_f$, and $s_{f_i}$ is the $i$-th element of the foreground frame prediction score sequence $S_f$, with $n$ being the length of sequence. The binary sequence $Y_f$ is constructed as follows: for a video feature sequence of length $n$, $a_s$ and $a_e$ denote the start and end boundary indices of the target segment, respectively. Frames within these indices are labeled as 1 (indicating foreground frames), while those outside are labeled as 0 (indicating background frames).

## A.3 Supervision of the Video-level Relevance Discriminator

The sentence-level query feature $h_Q$ and the query-related video representation $h_V$ are concatenated to produce a video-level relation signal vector $g \in \mathbb{R}^d$, represented as $g = \text{FC}([h_Q; h_V])$. To train the video-level relevance discriminator and ensure that the video-level relation signal vector accurately learns the semantic relationship between the query text and video segments, this paper introduces a video-level relation constraint $L_{video}$. Specifically, the video-level relation signal vector $g$ is input into a fully connected layer with a sigmoid function to obtain the video-level similarity score $score_{cg} = \text{sigmoid}(\text{FC}(g))$, which globally describes the relevance between the text and the video. The video-level relation constraint is calculated as:

$$L_{video} = -(y_h\log(score_{cg}) + (1 - y_h)\log(1 - score_{cg})),\tag{12}$$

where $y_h$ is the ground truth label indicating whether a sample has retrievable results or not.

## A.4 Relation-Aware Segment Predictor

Considering the video-level relation signal vector $g$ obtained from the video-level relevance discriminator contains enough relevance-aware knowledge to determine whether the segment exists or not, we utilize the video-level relation signal vector $g$ as the special token in this segment predictor. Specifically, using the foreground frame prediction score sequence $S_f$ produced by the Frame-level relevance discriminator, the foreground frames in the text-guided enhanced video feature $V_q$ are enhanced, while the background features unrelated to the query text are diminished. The foreground-enhanced video feature sequence $\tilde{V}_q$ is calculated as:

$$\tilde{V}_q = V_q \odot S_f,\tag{13}$$

where $\odot$ denotes element-wise multiplication. Subsequently, the video-level relation signal vector $g$ is concatenated to the front of the foreground-enhanced video feature sequence $\tilde{V}_q$, forming the input feature sequence $V_q^g = [g; \tilde{V}_q]$ for the relation-aware segment predictor, in dimension $\mathbb{R}^{(n+1)\times d}$. This sequence is then fed into a stacked two-layer LSTM network. The first layer of the LSTM generates the time-sequential feature sequence for predicting the start boundary $H^s = [h_0^s, h_1^s, \ldots, h_n^s]$ in $\mathbb{R}^{(n+1)\times d}$, and the second layer processes this output to generate the end boundary feature sequence $H^e = [h_0^e, h_1^e, \ldots, h_n^e] \in \mathbb{R}^{(n+1)\times d}$, as follows:

$$\begin{aligned} h_t^s &= \text{LSTM}_1(v_{qt}^g, h_{t-1}^s), \\ h_t^e &= \text{LSTM}_2(h_t^s, h_{t-1}^e), \end{aligned}\tag{14}$$

where $h_t^{s/e}$ denotes the $t$-th time-sequential feature vector in $H^{s/e}$, and $v_{qt}^g$ represents the $t$-th video feature vector in $V_q^g$. $\text{LSTM}_*$ indicates the corresponding layer of the LSTM network. The start and end time-sequential feature sequences $H^s$ and $H^e$ are each fed into their respective feedforward networks to obtain the boundary probability distributions $P_{s/e} \in \mathbb{R}^{(n+1)}$:

$$\begin{aligned} P_s &= \text{softmax}(FFN_{\text{start}}([H^s; V_q^g])), \\ P_e &= \text{softmax}(FFN_{\text{end}}([H^e; V_q^g])), \end{aligned}\tag{15}$$

where $P_s$ represents the start boundary probability distribution, and $P_e$ the end boundary probability distribution. The boundary prediction loss function $L_{boundary}$ is represented as:

$$L_{boundary} = -\frac{1}{2}\left(\sum Y_s\log(P_s) + \sum Y_e\log(P_e)\right),\tag{16}$$

where $Y_s$ and $Y_e$ are the one-hot vector representations of the ground-truth start ($a_s$) and end ($a_e$) boundary indices, respectively. $P_s$ and $P_e$ indicate the predicted start and end boundary probability distribution, respectively.

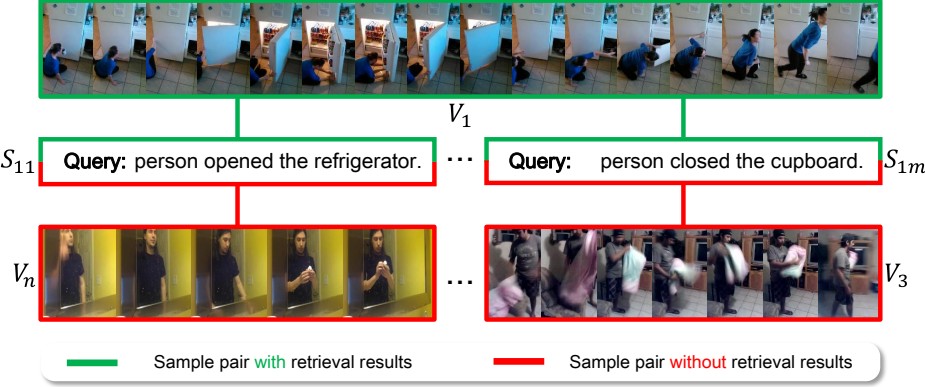

Figure 6: The organizational structure of reconstructed dataset.

Table 7: The detailed statistics of the different datasets.

| Dataset | Domain | #Videos | #Anno. | #Anno-RF. |
|---|---|---|---|---|
| Charades-STA [22] | Indoor | 6,672 | 12,408/1,860/1,860 | - |
| ActivityNet Captions [23] | Open | 19,207 | 37,421/17,505/17,031 | - |
| Charades-RF | Indoor | 6,672 | 12,408/3,720/3,720 | -/1,860/1,860 |
| ActivityNet-RF | Open | 19,207 | 37,421/35,010/34,062 | -/17,505/17,031 |

1 #Anno. denotes the number of video-text annotation pairs in different sets (train/val/test).
2 #Anno-RF. denotes the number of video-text annotation pairs without retrieval results in different sets (train/val/test).
3 The original Charades-STA dataset lacked a validation set, so we randomly split the test set into two equal parts to create a validation set and a new test set

## A.5  Dataset Reconstruction

Due to the absence of datasets specifically created for the TSG-RF task, this paper has reconstructed the validation and test sets of two widely used datasets in the TSG domain: Charades-STA [22] and ActivityNet Captions [23], to construct a testing environment for this task. In the original datasets, a single video corresponds to multiple query texts, as illustrated in Figure 6, where video $V_1$ corresponds to $m$ query texts with grounding results $[S_{11}, S_{12}, \ldots, S_{1m}]$. For each text with grounding results, a corresponding sample without grounding results (i.e., the video does not contain segments relevant to the query text) is constructed.

Specifically, the reconstruction process is exemplified by the sample pair $(S_{11}, V_1)$. By randomly selecting another video $V_n$ from the video library to pair with the query text $S_{11}$, a sample pair without grounding results $(S_{11}, V_n)$ is formed. To ensure the quality of reconstructed sample pairs and remove low-quality ones, this paper utilizes Large Language Models [62–64] for their deep and precise text modeling capabilities to achieve sample selection. Specifically, the BERT model [62] is used to extract features from the query text. Subsequently, the cosine similarity between the query text $S_{11}$ and the query texts with grounding results from the randomly selected video $[S_{n1}, \ldots, S_{nm}]$ is calculated. If the maximum similarity is less than a threshold value (empirically set to 0.2), the reconstructed sample pair without grounding results $(S_{11}, V_n)$ is added to the dataset. Otherwise, a new video is randomly selected until the condition is met. This methodology ensures the relevance and integrity of the dataset for accurately simulating the TSG-RF task.

As shown in Table 7, we name the corresponding reconstructed datasets as Charades-RF and ActivityNet-RF for the TSG-RF task. In these reconstructed datasets, we added an equal number of samples without grounding results to the validation and test sets. This augmentation ensures that the datasets more accurately reflect real-world scenarios where not all query texts correspond to relevant video segments.

## A.6  Implementation Details

Each word in the query text is initialized using GloVe300d, which remains frozen during training. Visual features of videos are extracted using a pre-trained I3D network. The maximum video feature sequence length is set to 128. Sequences longer than this are uniformly downsampled to 128, while shorter sequences are zero-padded to the same length. During training, since the training set was not reconstructed and only contains samples with

grounding results, each batch includes randomly selected videos paired with the original query text to create samples without grounding results. In Equation 3, we empirically set $\beta = 6$ and $\gamma = 6$ to balance all loss functions at the start of training. For the threshold $m$ in Equation 4, the value providing the highest accuracy on the validation set is chosen: 0.5 for Charades-RF and 0.3 for ActivityNet-RF. All experiments are conducted on a workstation with an NVIDIA GeForce RTX 3090Ti GPU and 256G RAM. Training our proposed model on the Charades-RF dataset takes approximately 2 hours, while training it on the ActivityNet-RF dataset takes approximately 5 hours.

## B  More Experiments

### B.1  Implement of the Enhanced Baseline with Relevance Feedback

Since there are currently no models specifically designed for the TSG-RF task, we select representative open-source models developed for the TSG task in recent years as baseline models to ensure fair and replicable comparisons, including VSLNet [13], SeqPAN [14], EAMAT [15], ADPN [16], UniVTG [49], QD-DETR [48]. While these baseline models demonstrate strong video segment grounding capabilities, they lack the ability to discern the relevance between the video and text in input samples, and thus assume that all samples contain retrievable results. Considering that directly training an additional relevance determination model is the most straightforward and efficient strategy to enable the above baseline models to discern relevance, this paper constructs a simple relevance determination model. By integrating this independently trained relevance determination model with the aforementioned baseline models, we compensate for their lack of relevance discernment capability, forming enhanced baseline models: VSLNet$^{++}$, SeqPAN$^{++}$, EAMAT$^{++}$, ADPN$^{++}$, UniVTG$^{++}$, and QD-DETR$^{++}$. Figure 7 illustrates the architecture of the simple relevance discrimination model. Specifically, the query text and video are processed through separate feature encoders. The encoded features of text and video are then concatenated along the sequence length dimension, with a CLS token added at the beginning to aid in learning the relevance between text and video. This concatenated sequence is subsequently input into a Transformer encoder to facilitate the interaction of information between feature vectors. Finally, the CLS token from the input sequence is processed through a fully connected layer to obtain the case discrimination score $P_v$.

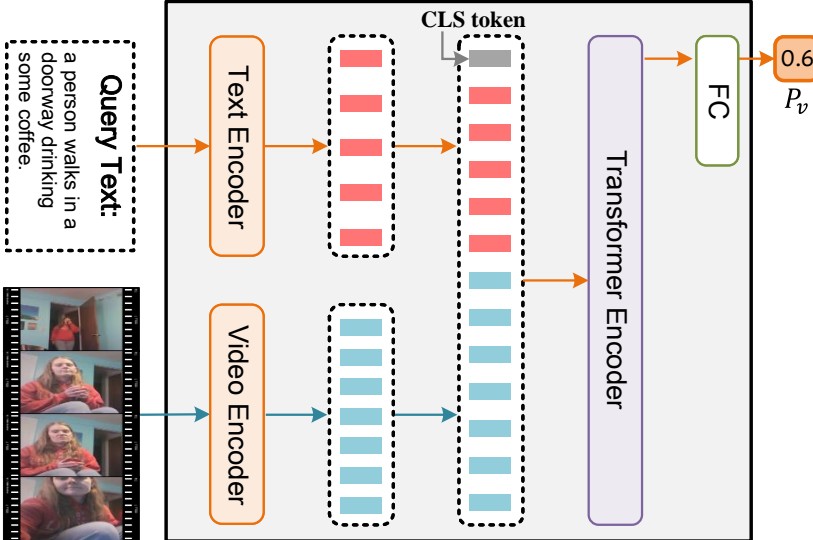

Figure 7: The architecture of the simple relevance discrimination Model.

### B.2  Grouped Performance Comparison with Baseline Methods

To further explore model performance under unbalanced data conditions, the number of samples without grounding results in the test set was randomly decreased in proportion, yielding six groups of test sets with ratios of non-grounding to retrieval samples at 0, 0.2, 0.4, 0.6, 0.8, and 1.0, respectively. Observing Figure 8 from left to right, when the proportion of non-grounding samples is 0, the test set contains only grounding-result samples, testing all models on their TSG task performance. Starting from a 0.2 ratio, all models are assessed on their TSG-RF task performance. As the proportion of non-grounding samples in the test set increases, although there is a general improvement in the mIoU of all models, the increase exhibited by RaTSG is the most significant. This is attributed to RaTSG's multi-scale learning of the semantic relationships between text and video, effectively integrating relevance feedback during the segment grounding process. Particularly in test sets

with a high proportion of non-grounding samples, RaTSG demonstrates superior performance compared to other models, showcasing its exceptional ability in case discrimination and fine-grained grounding.

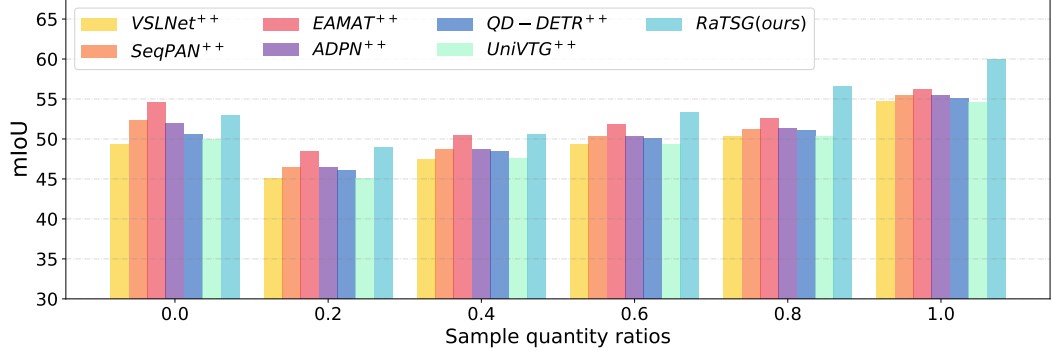

Figure 8: Performance Comparison across different models at varying sample quantity ratios.

## B.3 Performance Comparison in the Context of TSG Task

As illustrated in Table 8, we present a performance comparison of various models on the Charades-STA and ActivityNet Captions datasets for the TSG task. While our model, RaTSG, does not outperform all other models in every metric, it remains highly competitive. On the Charades-STA dataset, RaTSG achieves an R1@0.3 of 74.19%, which is very close to the top-performing EAMAT at 74.25%. For R1@0.5, RaTSG scores 56.61%, just slightly behind SeqPAN's 59.14%. In terms of mIoU, RaTSG records 53.02%, comparable to ADPN's 51.96%. Similarly, on the ActivityNet Captions dataset, RaTSG attains an R1@0.3 of 61.46%, nearly matching ADPN's 61.46% and EAMAT's 63.20%. For R1@0.5, RaTSG scores 42.36%, close to SeqPAN's 45.31%. In terms of mIoU, RaTSG achieves 43.72%, demonstrating its competitiveness alongside other models. Although RaTSG is not consistently the highest performer, it demonstrates competitive results across various metrics. It is important to note that our model is specifically designed for the TSG-RF task, which includes the challenge of handling samples without grounding results. This specialized focus may slightly affect its performance on the standard TSG task, yet RaTSG remains highly competitive. This demonstrates the robustness and versatility of our approach, indicating its strong potential in more complex real-world scenarios.

Table 8: Comparison on Charades-STA and ActivityNet Captions in the context of TSG task.

| Method | Charades-STA | | | | ActivityNet Captions | | | |
|---|---|---|---|---|---|---|---|---|
| | R1@0.3 | R1@0.5 | R1@0.7 | mIoU | R1@0.3 | R1@0.5 | R1@0.7 | mIoU |
| VSLNet | 67.47 | 54.62 | 35.43 | 49.37 | 62.12 | 43.76 | 25.64 | 44.54 |
| SeqPAN | 70.70 | 59.14 | 41.02 | 52.32 | 63.71 | 45.31 | 26.69 | 45.73 |
| EAMAT | 74.25 | 61.18 | 41.72 | 54.53 | 62.20 | 41.60 | 24.14 | 44.15 |
| ADPN | 71.24 | 56.88 | 39.73 | 51.96 | 61.46 | 41.49 | 24.78 | 44.12 |
| QD-DETR | 70.32 | 58.92 | 38.54 | 50.62 | 62.20 | 41.60 | 24.14 | 44.15 |
| UniVTG | 71.62 | 60.06 | 33.34 | 49.92 | 61.78 | 43.34 | 22.59 | 42.71 |
| RaTSG(ours) | 74.19 | 56.61 | 37.47 | 53.02 | 61.46 | 42.36 | 24.74 | 43.72 |

## B.4 Comparison to CLIP as relevance discriminator

As shown in Table 9, we also try to compare our designed relevance discriminator with large-scale pre-trained vision-language models due to their strong cross-modal representation capabilities. To implement the comparison, we choose the CLIP model, a large-scale pre-trained vision-language model, to assess the relevance between the query text and video. Specifically, we first utilize the CLIP to measure the cosine similarity score between the query text and each frame of a video, and then aggregate the similarity scores over all frames to obtain the final relevance. In particular, we use two aggregate methods implemented by the CLIP model: (1) averaging all scores (CLIP-Avg) and (2) averaging on top-5 scores (CLIP-Top5). We incorporate CLIP with the SOTA method EAMAT to achieve the joint relevance feedback and grounding framework. The experimental results on the Charades-RF dataset are summarized in Table 9. We can find that CLIP-based EAMAT models achieve relatively lower performance, demonstrating that the CLIP model has a poor ability to predict relevance. We attribute it

to the fact that the CLIP model is simply pre-trained on common scenarios without additional fine-tuning on specific downstream task, thus severely suffering from domain shift issues on the target dataset.

Table 9: Comparison of relevance discrimination ability with CLIP on Charades-RF dataset.

| Method | Acc | R1@0.3 | R1@0.5 | R1@0.7 | mIoU |
|---|---|---|---|---|---|
| EAMAT$^{++}$ | 71.94 | 63.55 | 59.17 | 51.96 | 56.23 |
| EAMAT$^{CLIP-Avg}$ | 60.70 | 54.92 | 51.96 | 47.10 | 50.21 |
| EAMAT$^{CLIP-Top5}$ | 62.52 | 55.75 | 52.47 | 46.94 | 50.25 |
| **RaTSG(Ours)** | **76.85** | **68.17** | **61.91** | **54.19** | **59.93** |

