# OpenReview forum: "Temporal Sentence Grounding with Relevance Feedback in Videos"
_NeurIPS.cc/2024/Conference — NeurIPS 2024 poster_

### Official Review · Reviewer_P4nK · 2024-07-09

**Soundness:** 4
**Presentation:** 3
**Contribution:** 4
**Rating:** 8
**Confidence:** 5

**Summary:**

This paper proposes a new task called Temporal Sentence Grounding with Relevance Feedback (TSG-RF) in videos, which extends the traditional Temporal Sentence Grounding (TSG) by introducing cross-modal video-text semantic relevance prediction. Besides, a new Relation-aware Temporal Sentence Grounding (RaTSG) network is proposed, where a multi-granularity relevance discriminator and a relation-aware segment grounding module are specifically devised for the TSG-RF. Quantitative and qualitative experiments conducted on two reconstructed datasets demonstrate the effectiveness of our proposed RaTSG for the new TSG-RF task.

**Strengths:**

1.	Compared to the traditional TSG, the proposed new TSG-RF task is interesting and more in line with practical application scenarios. Besides, two TSG datasets are reconstructed to make them suitable for evaluating the TSG-RF methods. The TSG-RF task with corresponding benchmarks, holds significant value for the vision-language community.
2.	The proposed multi-granularity relevance discriminator and the relation-aware segment grounding module are simple and effective. It has been demonstrated that they mutually enhance each other.
3.	The experimental results are convincing. Although there are no existing methods specifically designed for TSG-RF, the paper compares the recent TSG models and their extending variants for TSG-RF. The ablation experiments and visualization results demonstrate the effectiveness of the proposed method.
4.	The writing of the paper is easy to follow. The source code and dataset are released.

**Weaknesses:**

The paper should illustrate bad examples in the visualization section, and discuss the limitations of the proposed method.

**Questions:**

1.	The proposed relation-aware segment predictor is key for TSG-RF. Could it be adapted for traditional TSG methods to enable them to work for TSG-RF?
2.	As the proposed method requires additional relevance feedback, I wonder how the extra computational workload would be included.

**Limitations:**

Although the proposed method could perform extra relevance feedback, its complexity may be increased when compared to traditional TSG methods.

---

> ### Author Rebuttal · Authors · 2024-08-07
>
> Dear Reviewer P4nK,
>
> Thank you for your comprehensive and positive review of our work. We appreciate your insights and suggestions for further improvement. Below, we address your concerns point by point.
>
> **Q1: The paper should illustrate bad examples in the visualization section, and discuss the limitations of the proposed method.**
>
> **A1**:  Following the reviewer's suggestion, we have included two bad examples in the attached PDF file of the global response. In the first example of the query text  "A person starts sneezing", our model incorrectly judges the relevance feedback due to the lack of audio cues, which are crucial for identifying the action of sneezing. In the second example, the query text is "A person opening a laptop," but the video segment actually shows the person closing a laptop. Our model misinterprets the temporal sequence of actions, mistaking the closing action for the opening action. These examples demonstrate that our proposed model struggles with handling audio-related actions and temproal-sensitive content. However, such limitation can be alleviated by integrating audio features and temporal modeling in the video representation moudel.
>
>
> **Q2: The proposed relation-aware segment predictor is key for TSG-RF. Could it be adapted for traditional TSG methods to enable them to work for TSG-RF?**
>
> **A2**:  Of course, we can adapte our main components to traditional TSG methods to enable them to work for TSG-RF. Note that, to effectively address the TSG-RF task, it is essential to use not only the Relation-aware Segment Predictor (RaSP) but also the Multi-granularity Relevance Discriminator (MgRD). These components work together to assess relevance at multiple granularities and provide accurate segment grounding. Following the reviewer's suggestion, we adapt them into an existing traditional TSG model EAMAT, which is the best performer among our four compared models. As shown in the following table, EAMAT with our devised RaSP and MgRD outperfomes the original EAMAT and the enhanced EAMAT++ with  an extra relevance discriminator. The results demonstrate the adaptability and effectiveness of our proposed modules for enhancing traditional TSG methods to work for the TSG-RF task.
>
> | Model | Acc | R1\@0.3 |R1\@0.5|R1\@0.7| mIoU |
> |------------|----|----------------|-----------------|----------------|-------|
> | EAMAT | 50.00 | 37.12 | 30.59 | 20.86 | 27.27 |
> | EAMAT++ | 71.94 | 63.55 | 59.17 | 51.96 | 56.23 |
> | **EAMAT+Ours** | **76.37** | **67.47** | **62.02** | **54.83** | **59.58** |
>
> **Q3: As the proposed method requires additional relevance feedback, I wonder how the extra computational workload would be included.**
>
> **A3**:  To measure model's computational workload, we use GFLOPs and parameter count. By adding relevance feedback capability to our model, the GFLOPs and parameter count increase from 0.077 to 0.081 G  and 1.21 to 1.27 M , respectively, with a relatively small increase of 5.1% and 4.9%. This negligible increase is due to our multi-task framework, where the additional relevance feedback module shares the feature extraction modules with the segment grounding module. This efficient design ensures that the overall computational workload remains low while enhancing the model with relevance feedback capability.
>
> We will incorporate all the above in the final version. Thank you.

---

> > ### Comment · Reviewer_P4nK · 2024-08-13
> > **Post response**
> >
> > My concerns have been addressed. Thanks for the effort from the authors. Lastly, I recommend including Figure 4 by adding a bad example in the revision, and adding the results of Q2 in the appendix. Considering that the paper presents a new interesting and practical task, introduces a simple yet effective method, and provides informative open-source code, I would like to increase my final rating as a strong accept.

---

> > > ### Author Response · Authors · 2024-08-13
> > > **Thanks very mauch. We will update our paper accordlingly.**
> > >
> > > Thanks for your suggestion, we will include Figure 4 by adding a bad example and updating the discussion accordingly in the revision, and will add the results of Q2 in the appendix. Thanks again for your positive review.

---

### Official Review · Reviewer_eoKJ · 2024-07-10

**Soundness:** 4
**Presentation:** 4
**Contribution:** 4
**Rating:** 7
**Confidence:** 5

**Summary:**

This paper presents a novel task named Temporal Sentence Grounding with Relevance Feedback (TSG-RF) to overcome the limitations of conventional Temporal Sentence Grounding (TSG), which presumes the existence of relevant segments in videos. This paper introduces the Relation-aware Temporal Sentence Grounding (RaTSG) network, incorporating a multi-granularity relevance discriminator and a relation-aware segment grounding module to generate relevance feedback and segment boundaries. The RaTSG network's efficacy is demonstrated by reconstructing two widely-used datasets for TSG-RF and validating its performance through extensive experiments. Additionally, the source code for RaTSG is made available.

**Strengths:**

1.This paper introduces the TSG-RF task, which addresses the limitations of traditional TSG by considering scenarios where query-related segments may be absent in videos.
2.The proposed RaTSG network utilizes a multi-granularity relevance discriminator and a relation-aware segment grounding module, showing significant performance improvements and achieving state-of-the-art results in the TSG-RF task.
3.The experimental validation is comprehensive, featuring reconstructed datasets and extensive experiments that clearly demonstrate the effectiveness of the RaTSG network. The results are presented in a clear and well-organized manner.

**Weaknesses:**

1.The weaknesses of this paper include a lack of sufficient baselines for comparison with the RaTSG network. For instance, the paper does not investigate the use of Large Multimodal Models (LMMs) to assess the relevance between the query text and video, which could offer a more robust benchmark for evaluating the proposed method. Incorporating such comparisons would enhance the overall evaluation of the RaTSG network.
2.The paper also lacks adequate qualitative analysis. Including more qualitative examples of both successful and unsuccessful cases would provide deeper insights into the strengths and limitations of the RaTSG network.

**Questions:**

Please refer to the weakness.

**Limitations:**

The paper does not discuss the limitations of the work. However, the introduction of the novel TSG-RF task significantly improves upon the limitations of traditional TSG task, offering positive societal impact. For constructive suggestions on areas needing improvement, please refer to the weaknesses.

---

> ### Author Rebuttal · Authors · 2024-08-07
>
> Dear Reviewer eoKJ,
>
> Thank you for your detailed and positive review of our work. We appreciate your insights and suggestions for further improvement. Below, we address your concerns point by point.
>
> **Q1: The weaknesses of this paper include a lack of sufficient baselines for comparison with the RaTSG network. For instance, the paper does not investigate the use of Large Multimodal Models (LMMs) to assess the relevance between the query text and video, which could offer a more robust benchmark for evaluating the proposed method. Incorporating such comparisons would enhance the overall evaluation of the RaTSG network.**
>
> **A1**: While we acknowledge the importance of comparing our method with large multimodal models (LMMs), it is important to note that current state-of-the-art LMMs, such as Llava and GPT-4, do not yet have the capability for tackling the video-based task. Instead, we choose  a large-scale pre-trained vision-language model, i.e., CLIP, to assess the relevance between the query text and video. Specifically, CLIP is utilzed to measure the cosine similarity score between the query text and each frames of a video, and the final relevance is obtained by aggregating the similarity scores over all frames. We use two aggregate methods: averaging all scores (CLIP-Avg) and  averaging on top-5 scores (CLIP-Top5). Then we jointly use CLIP and  EAMAT that is the best performer among our four compared models, to conduct relevance feedback and grouding. The reulsts on Charades-RF are summarized in the following table. CLIP-based models achieve low accuracy, demonstating its poor ability for relevance prediction. We attribute it to the fact that CLIP is pretrained without addtional training, thus suffering from domain shift issues on the traget dataset.
>
> | Model | Acc  | R1\@0.3 | R1\@0.5 | R1\@0.7 | mIoU  |
> |----------------|-----|--------|------    |--------|-------|
> | $EAMAT^{++}$ | 71.94 | 63.55   | 59.17  | 51.96  | 56.23 |
> | $EAMAT^{CLIP-Avg}$ |60.70  | 54.92   | 51.96  | 47.10  | 50.21 |
> | $EAMAT^{CLIP-Top5}$| 62.52| 55.75   | 52.47  | 46.94  | 50.25 |
> | **RaTSG (ours)** | 76.85| 68.17  | 61.91 | 54.19 | 59.93 |59.93 |
>
>
> **Q2: The paper also lacks adequate qualitative analysis. Including more qualitative examples of both successful and unsuccessful cases would provide deeper insights into the strengths and limitations of the RaTSG network.**
>
> **A2**:  To discuss the limitations of the work, we have included two unsuccessful examples in the attached PDF file of the global response. In the first example of the query text  "A person starts sneezing", our model incorrectly judges the relevance feedback due to the lack of audio cues, which are crucial for identifying the action of sneezing. In the second example, the query text is "A person opening a laptop," but the video segment actually shows the person closing a laptop. Our model misinterprets the temporal sequence of actions, mistaking the closing action for the opening action. These examples demonstrate that our proposed model struggles with handling audio-related actions and temproal-sensitive content. However, such limitation can be alleviated by integrating audio features and temporal modeling in the video representation moudel. For the successful examples and corresponding discussion, we illustrate them in Figure 4 and Section 4.4.
>
> We will incorporate all the above in the final version. Thank you.

---

> > ### Comment · Reviewer_eoKJ · 2024-08-13
> >
> > I have read the other reviewer's comments and author's reply, and consider the paper meet the standards required for NeurIPS. I recommend accepting the paper.

---

> > > ### Author Response · Authors · 2024-08-13
> > > **Thanks for your recommendation.**
> > >
> > > Thanks for your time, and we greatly appreciate your recommendation on our paper.

---

### Official Review · Reviewer_dpih · 2024-07-11

**Soundness:** 2
**Presentation:** 2
**Contribution:** 2
**Rating:** 5
**Confidence:** 5

**Summary:**

This paper introduces Temporal Sentence Grounding with Relevance Feedback (TSG-RF) in videos, a new task that addresses the limitations of traditional Temporal Sentence Grounding (TSG), which assumes relevant segments always exist within a video. TSG-RF accounts for the possibility that a video may not include a segment related to the query, aiming to localize segments that align with the query when present and provide feedback when they are absent. The proposed Relation-aware Temporal Sentence Grounding (RaTSG) network reformulates TSG-RF as a foreground-background detection problem, assessing query-related semantics at both frame and video levels. It utilizes a multi-granularity relevance discriminator for precise relevance feedback and a relation-aware segment grounding module to adaptively ground segments. To validate RaTSG, two popular TSG datasets are reconstructed, establishing a benchmark for TSG-RF. Experimental results demonstrate the effectiveness of RaTSG for this task.

**Strengths:**

- This paper introduces the Relation-aware Temporal Sentence Grounding (RaTSG) network, which effectively uses a multi-granularity relevance discriminator for predicting relevance feedback and a relation-aware segment grounding module for selectively determining segment boundaries.
- The paper argument the original datasets for evaluation.

**Weaknesses:**

- The key motivation of this paper is related to another video temporal task -- highlight detection. However, this paper do not have any discussion.
- The method is cumbersome, featuring several incremental designs.
- The experiments utilize outdated datasets. Newer datasets, such as QV-Highlights, should be considered.
- The baselines are no up to date which do not include sota methods. such as UMT, UniVTG, QD-DETR.

**Questions:**

Please refer to the Weaknesses section.

**Limitations:**

The Relevance feedback in Video Moment Retrieval is a trivial research question. Either we could generalize this problem to general video grounding (spatial / temporal) or general video understanding (not just VMR, but also hallucinations in VLMs).

---

> ### Author Rebuttal · Authors · 2024-08-07
>
> Dear Reviewer dpih,
>
> Thanks for your detailed review and constructive feedback. We appreciate your insights and would like to address your concerns point by point.
>
>
> **Q1:  discussion about highlight detection**
>
> **A1**: We thank the reviewer for sharing the insight. Actually, our RF task is quite different from highlight detection. Specifically, highlight detection mainly aims to identify the most interesting or important segments within a video based on a given natural language query, focusing on segments that are salient or engaging. This task is similar to the traditional temporal sentence grounding, and existing highlight detection methods still naively assume that there are always highlights or interesting segments in the video. In contrast, our RF setting is specifically designed to handle cases where no relevant segments are found. This involves providing relevance feedback to indicate when no relevant segments exist, which is essential for practical applications where not all queries have corresponding relevant segments. We believe that our RF setting can also be plug-and-play to be applied to the highlight moment detection tasks. We will discuss the  highlight detection in the related work.
>
> **Q2: The method is cumbersome, featuring several incremental designs.**
>
> **A2**: Our proposed framework is specifically designed to handle the challenges posed by our new introduced  TSG-RF task, incorporating several new key modules to ensure effective performance (**please see the global response**). Moreover, our RaTSG framework is a multi-task model that achieves high efficiency with relatively low complexity. As shown in Table 1 of our paper, our model has fewer parameters compared to other baseline models, indicating that our approach is not cumbersome.
>
> **Q3: The experiments utilize outdated datasets. Newer datasets, such as QV-Highlights, should be considered.**
>
> **A3**: As shown in the table below, we make a statistics on the number of top-tier conferrence papers (NeurIPS、CVPR、ICCV、ECCV、AAAI、ACM MM) using the three datasets over the past three years, finding that Charades-STA and ActivityNet Captions are more popular than QV-Highlights. Though including QV-Highlights would enhance our work, we believe that the extensive experiments on two widely-used TSG datasets are convincing. Moreover, we try to use QV-Highlights, but we could not complete dataset downloading, reconstruction, and running experiments with multiple baselines within the limited rebuttal time. We leave the use of QV-Highlights for future work.
>
> | Dataset Name | 2021 | 2022 | 2023 |
> |--------------|----- |-----|------|
> | Charades-STA | 10 | 12 | 19 |
> | ActivityNet Captions | 12 | 13 | 14 |
> | QV-Highlights | 1 | 1 | 6 |
>
> **Q4: The baselines are no up to date which do not include sota methods. such as UMT, UniVTG, QD-DETR.**
>
> **A4**: We would like to clarify that our paper does include comparisons with recent state-of-the-art methods:
>
> - The following table summarizes the traditional TSG performance comparison between the suggested UMT, UniVTG, QD-DERT and our compared ADPN on Charades-STA. It shows that our chosen ADPN is almost the SOTA method, performing much better than the other methods, especially in terms of R1\@0.7. Therefore, we argue that our compared method is not outdated.
>
> | Method| R1\@0.3 | R1\@0.5 | R1\@0.7 | mIoU|
> |----|---|----|---|---|
> | UMT (2022) | -| 48.31  | 29.25  | - |
> | UniVTG (2023)  | 70.81  | **58.01**  | 35.65  | 50.10 |
> | QD-DETR (2023) | -| 57.31  | 32.55  | - |
> | **ADPN (2023)**|**71.99**|57.69|**41.10**|**52.86**|
>
> - Additionally, following the reviewer's suggestion, we have included a comparison with UniVTG and QD-DETR. Note that we did not compare with UMT considering its performance is much worse than the others (please see the above table). The table below summarizes the performance on the Charades-RF dataset. Our proposed model still outperforms the UniVTG and QD-DETR counterparts. We will  update Table 1 by including UniVTG and QD-DETR and revise the corresponding discussion accordingly in the revison.
>
> | Model | Acc|R1\@0.3 |R1\@0.5 |R1\@0.7 | mIoU |
> |------|----|-----|----|---|------|
> | UniVTG | 50.00| 35.81| 30.03| 16.67| 24.96 |
> | QD-DETR | 50.00| 35.16| 29.46| 19.27| 25.31 |
> | UniVTG++ | 71.94|62.58| 58.55| 48.79| 54.65 |
> | QD-DETR++| 71.94| 62.18| 58.20| 50.96| 55.13 |
> | **RaTSG (ours)** | **76.85**| **68.17**| **61.91**| **54.19**| **59.93**|
>
> **Q5: The Relevance feedback in Video Moment Retrieval is a trivial research question. Either we could generalize this problem to general video-related tasks.**
>
> **A5**: We respectfully disagree with the assertion that relevance feedback in video moment retrieval is a trivial research question. Actually, the TSG-RF task addresses a significant gap in the field of temporal sentence grounding by introducing a scenario where no relevant segments might exist in the video. This is a critical aspect that current methods do not adequately address. The relevance feedback mechanism in our work is crucial for several reasons:
>
> - **More Practical**: In real-world applications, not every query will have a relevant segment in the video. Providing accurate feedback on the absence of relevant segments is essential for practical usability.
>
> - **Enhanced Accuracy**: The feedback mechanism improves the precision and reliability of the grounding process by dynamically adjusting based on the presence or absence of relevant segments.
>
> - **Potential for Generalization**: Our framework has the potential to be generalized to broader video grounding tasks, including spatial and temporal grounding, as well as general video understanding tasks. The principles of relevance feedback can be adapted to various contexts, enhancing the robustness and versatility of grounding models across different tasks.
>
> We will incorporate all the above in the final version (as specified in our response). Please reconsider our paper. Thank you.

---

> > ### Comment · Reviewer_dpih · 2024-08-12
> > **Post response**
> >
> > Thank you to the authors for their rebuttal and hard work.
> >
> > Regarding Q1 on highlight detection: The explanation that RF setting is specifically designed to handle cases where no relevant segments are found makes sense.
> >
> > For Q2: QVHL is distinct because it includes one or more ground-truth intervals with high-quality resolutions, making it more reliable and challenging compared to Charades-STA and ActivityNet, which many past methods have already overfitted on.
> >
> > I appreciate the addition of several baseline comparisons in the experiments. Please ensure the discussion on highlight detection is included in the related work and add the baseline (UMT, UniVTG, etc) comparison in the experiments, as omitting this would be disrespectful to the work in this community.
> >
> > Considering these points, I would like to increase my rating to borderline accept.

---

> > > ### Author Response · Authors · 2024-08-13
> > > **Thanks for you suggestion, and we will update our paper accordlingly.**
> > >
> > > Many thanks for sharing the insight with us. We will utilize QV-Highlights in our follow-up works. Additionally, we promise that we will incorporate the discussion on highlight detection and appropriately cite relevant papers in the related work. Besides, we will update Table 1 by including UniVTG and QD-DETR for a more comprehensive comparison.

---

> > > > ### Comment · Reviewer_dpih · 2024-08-14
> > > > **Post response**
> > > >
> > > > Thanks for the response, the author have addressed my concerns.

---

### Official Review · Reviewer_3Qyt · 2024-07-12

**Soundness:** 3
**Presentation:** 2
**Contribution:** 1
**Rating:** 5
**Confidence:** 5

**Summary:**

The work develops a model that localizes query-related segments when present and provides feedback on non-existence when absent.
It proposes a Relation-aware Temporal Sentence Grounding (RaTSG) network, which reformulates TSG-RF as a foreground-background detection problem.
Also, the work uses a multi-granularity relevance discriminator for precise video-query relevance feedback.
The work constructs two popular TSG datasets for TSG-RF benchmarking.
Experimental results demonstrate RaTSG's effectiveness for the TSG-RF task

**Strengths:**

The work introduces a new and more realistic task, TSG-RF, which addresses a significant limitation of traditional TSG.
The work develops a model that can handle cases where no relevant segment exists, making it more suitable for real-world applications.
Experimental results demonstrate the effectiveness of the proposed RaTSG network for the TSG-RF task.

**Weaknesses:**

1. The work claimed to incorporate foreground/background info. However, it was simply adding a binary classification, such binary foreground classification for temporal grounding was already explored in [1]. The video part is done by weighted sum, which is also a standard way of aggregating frame features.
2. In Table 1, the author explained why the performance is 50%. Why are the other rows all have 71.94% of accuracy and 81.6% accuracy?
3. Although this work proposes a new RF setting, it will be interesting to know how the model performs in a standard setting to see if there is a performance gap.



[1] Lin, Kevin Qinghong, et al. "Univtg: Towards unified video-language temporal grounding." Proceedings of the IEEE/CVF International Conference on Computer Vision. 2023.

**Questions:**

Please address the questions in weakness.

---

> ### Author Rebuttal · Authors · 2024-08-07
>
> Dear Reviewer 3Qyt,
>
> Thank you for your thorough review and valuable feedback. We appreciate your insights and would like to address your concerns point by point.
>
> **Q1:  The work claimed to incorporate foreground/background info. However, it was simply adding a binary classification, such binary foreground classification for temporal grounding was already explored in [1]. The video part is done by weighted sum, which is also a standard way of aggregating frame features.**
>
> **A1**:  In our work, we make the first attempt to fill the critical gap in traditional temporal sentence grounding tasks that assume the existence of relevant segments in the corresponding video but fails to provid feedback on non-existence relevant content. In addition to proposing a novel TSG-RF task and a new framework, we claim that we have also developed novel technical designs for the pipeline.
>
> In particular, as for the foreground/background information learning, our foreground-background detection is not simply implemented with a binary classification. Although the naive binary classification head has been investigated and utilized to detect the foreground clips in previous work [1], it is simply/directly applied to the raw clip-level features for classification. Such design is coarse and fail to capture the long-term dependency in the complex videos. In contrast, we propose a more effective multi-granularity relevance discriminator. By employing both frame-level and video-level relevance discriminators, our dual-level multi-granularity relevance discriminator allows us to effectively balance fine-grained and coarse-grained relevance assessments, thereby enhancing the overall robustness and applicability of our model.
>
> Additionally, while the weighted sum operation is indeed a standard operation in video analysis, it is effective enough to handle the frame-wise contexts aggregation for determining the query-related video-level content. Moreover, aggregating frame features is not the focus of our work, and any feature aggregation operation can be theoretically used in our proposed framework. Note that our main contribution is proposing a novel TSG-RF task and a new framework that incorporates relevance feedback capability.
>
>
> **Q2:  In Table 1, the author explained why the performance is 50%. Why are the other rows all have 71.94% of accuracy and 81.6% accuracy?**
>
> **A2**:  In Table 1, the performance of traditional TSG methods, such as VSLNet, shows 50% accuracy because our newly constructed test set has an equal ratio (1:1) of samples with and without grounding results (i.e., query-relevant segments exist or do not exist in the corresponding video). Since the traditional TSG methods assume that all samples have grounding results, they lead to a relevance prediction accuracy of 50%.
>
> For the enhanced versions of the traditional models (i.e., VSLNet++, ADPN++, SeqPAN++,  and EAMAT++) , we incorporate an additional relevance discriminator to their models. This relevance discriminator is trained separately and then directly combined with the baseline methods, endowing them with the same ability to discriminate relevance (Please  refer to Appendix B.1 for more implementation details). This additional relevance discriminator achieves the 71.94% and 81.6% accuracy on Charades-RF and ActivityNet-RF, respectively. It is worth noting that as all enhanced models share the same relevance discriminator, they achive the same accuracy on the same dataset.
>
> **Q3: Although this work proposes a new RF setting, it will be interesting to know how the model performs in a standard setting to see if there is a performance gap.**
>
> **A3**:  Actually we have conducted experimetns with the standard setting in Appendix B.3 of our paper. Below is the Table  from the appendix, summarizing the performance comparison in the context of the TSG task. Although our proposed RaTSG is not consistently the highest performer, it demonstrates competitive results across various metrics. It is important to note that our model is specifically designed for the TSG-RF task, which includes the challenge of handling samples without grounding results. This specialized focus may slightly affect its performance on the standard TSG task, yet RaTSG remains highly competitive. This demonstrates the robustness and versatility of our approach, indicating its strong potential in more complex real-world scenarios.
>
> | Method| Charades-STA (R1\@0.3) | Charades-STA (R1\@0.5) | Charades-STA (R1\@0.7) | Charades-STA (mIoU) | ActivityNet (R1\@0.3) | ActivityNet (R1\@0.5) | ActivityNet (R1\@0.7) | ActivityNet (mIoU)|
> |-----|-----|-----|-----|-----|-----|-----|-----|-----|
> | VSLNet | 67.47| 54.62| 35.43| 49.37| 62.12| 43.76| 25.64| 44.54|
> | SeqPAN | 70.70| 59.14| 41.02| 52.32| 63.71| 45.31| 26.69| 45.73|
> | EAMAT| 74.25| 61.18| 41.72| 54.53| 62.20| 41.60| 24.14| 44.15|
> | ADPN| 71.24| 56.88| 39.73| 51.96| 61.46| 41.49| 24.78| 44.12|
> | RaTSG (Ours) | 74.19| 56.61| 37.47| 53.02| 61.46| 42.36| 24.74| 43.72|
>
>
> We will incorporate all the above in the final version. Please reconsider our paper. Thank you.

---

> > ### Comment · Reviewer_3Qyt · 2024-08-12
> >
> > Thanks for the rebuttal!
> >
> > The response to Q2 and Q3 addressed my concerns. I'll raise my score accordingly.

---

> > > ### Author Response · Authors · 2024-08-13
> > > **Thank you for raising the score.**
> > >
> > > Thank you for raising the score. We also appreciate your detailed review with constructive feedback.

---

### Author Rebuttal · Authors · 2024-08-07

We thank all reviewers for their encouragement and guidance to further improve this work. **Here, we would like to discuss the specific design of our RaTSG framework for TSG-RF task.**

Firstly, we would like to emphasize that the primary innovation of our paper is the introduction of the new and challenging TSG-RF task. This is the first time this task has been proposed, addressing the critical gap in traditional temporal sentence grounding tasks that assume the existence of relevant segments in every video.

Secondly, to tackle this novel TSG-RF task, we propose a new Relation-aware Temporal Sentence Grounding (RaTSG) framework. Our framework is specifically designed to handle the challenges posed by this task, incorporating several new key modules to ensure effective performance:

- **Multi-granularity Relevance Discriminator**:
The multi-granularity relevance discriminator addresses the challenge of relevance feedback by evaluating the relevance between the query text and video at multiple granularities.
   - **Frame-level relevance discriminator**:  It evaluates the relevance of each video frame with the query text on a fine-grained basis, capturing detailed and nuanced relationships. This ensures a precise measurement of relevance for each frame.
   - **Video-level relevance discriminator**: It aggregates these frame-level relevance scores using a weighted sum approach, providing a broader context by calculating the relative relevance of each frame within the entire video sequence.

- **Dynamic Relevance Feedback Mechanism**:
The dynamic relevance feedback mechanism is designed in the relation-aware segment grounding to handle the dynamic nature of relevance feedback. This mechanism dynamically adapts the segment grounding process based on the relevance feedback received. By iteratively refining predictions, the mechanism improves accuracy and ensures the model can handle cases where no relevant segment exists. The feedback integration enhances the model's ability to adapt to varying levels of relevance, ensuring practical usability.

- **Mutual Enhancement through Joint Training**:
Our RaTSG method employs a joint training approach that promotes mutual enhancement between relevance discrimination and segment grounding. By sharing representations and learning jointly, the model benefits from the complementary strengths of both tasks. Relevance discrimination improves the model's ability to identify relevant segments, while segment grounding enhances its understanding of contextual relationships. As demonstrated in Section 4.3 (Ablation Studies) of our paper, the joint training approach leads to significant performance improvements.

In summary, our RaTSG framework is carefully designed to address the specific challenges of the TSG-RF task. Each module plays a critical role in ensuring the model's robustness and accuracy in identifying relevant segments and providing relevance feedback when no relevant segments are found.

---

### Decision · Program_Chairs · 2024-09-25

**Decision:**

Accept (poster)

**Comment:**

Initially, the reviewers raise some questions regarding the way to incorporate foreground/background info, the model performance in a standard setting, comparison with another video temporal task-highlight detection, incremental designs, dataset and baseline usage, the lack of adequate qualitative analysis, etc. Most of these questions are addressed in the rebuttal and recognized by reviewers. Eventually with the merit of the work, this paper receives one strong accept, one accept, and two borderline accepts, with agreement for acceptance. The AC recommends to accept. Authors are encouraged to revise the paper according to the reviews.